# Metabolic Derangement of Essential Transition Metals and Potential Antioxidant Therapies

**DOI:** 10.3390/ijms25147880

**Published:** 2024-07-18

**Authors:** Adriana Fontes, Adrian T. Jauch, Judith Sailer, Jonas Engler, Anabela Marisa Azul, Hans Zischka

**Affiliations:** 1Institute of Molecular Toxicology and Pharmacology, Helmholtz Center Munich, German Research Center for Environmental Health, D-85764 Neuherberg, Germany; adrianafilipa.dasilvafontes@helmholtz-munich.de; 2CNC-Center for Neuroscience and Cell Biology, University of Coimbra, 3004-504 Coimbra, Portugal; 3CIBB-Center for Innovative Biomedicine and Biotechnology, University of Coimbra, 3004-504 Coimbra, Portugal; 4School of Medicine and Health, Institute of Toxicology and Environmental Hygiene, Technical University Munich, D-80802 Munich, Germany; 5IIIUC-Institute for Interdisciplinary Research, University of Coimbra, 3030-789 Coimbra, Portugal

**Keywords:** transition metals, iron, copper, zinc, manganese, oxidative stress, antioxidants

## Abstract

Essential transition metals have key roles in oxygen transport, neurotransmitter synthesis, nucleic acid repair, cellular structure maintenance and stability, oxidative phosphorylation, and metabolism. The balance between metal deficiency and excess is typically ensured by several extracellular and intracellular mechanisms involved in uptake, distribution, and excretion. However, provoked by either intrinsic or extrinsic factors, excess iron, zinc, copper, or manganese can lead to cellular damage upon chronic or acute exposure, frequently attributed to oxidative stress. Intracellularly, mitochondria are the organelles that require the tightest control concerning reactive oxygen species production, which inevitably leaves them to be one of the most vulnerable targets of metal toxicity. Current therapies to counteract metal overload are focused on chelators, which often cause secondary effects decreasing patients’ quality of life. New therapeutic options based on synthetic or natural antioxidants have proven positive effects against metal intoxication. In this review, we briefly address the cellular metabolism of transition metals, consequences of their overload, and current therapies, followed by their potential role in inducing oxidative stress and remedies thereof.

## 1. Introduction

The trace metals iron (Fe), copper (Cu), zinc (Zn), and manganese (Mn) are essential for cellular metabolism in living organisms [1], being involved in amino acid, lipid, protein, and carbohydrate metabolism [2,3]. They act as structural cofactors for proteins and nucleic acids, and are essential for oxygen transport, neurotransmitter synthesis, nucleic acid repair, and construction of the extracellular matrix [4], as well as cellular respiration and transcription [5]. Despite their importance, the transition metals Cu^2+/+^, Fe^3+/2+^, and Mn^3+/2+^ can participate in electron transfer reactions, which can result in damage to structural cellular components, potentially inducing cell death [1]. To this, intra and extracellular systems are equipped with different mechanisms that ensure their uptake, storage, and distribution to their respective targets.

Intoxications caused by transition metals may have their origins in air, water, or soil contamination and agriculture (namely, resulting from the use of fertilizers and pesticides). Other anthropogenic activities, like mining and smelting, can also be a source of metal intoxication in humans [6]. Genetic conditions [1] and aging [7,8,9,10] represent further circumstances that may contribute to the accumulation of transition metals in tissues such as liver and brain; the latter being of substantial concern with respect to neurodegenerative diseases [11,12,13]. Metal imbalances in the body have also been associated with conditions like cardiovascular disease [14,15,16,17], metabolic syndrome [18,19,20], cancer [21,22,23], inflammatory bowel disease [24,25,26], and even depression and anxiety [27,28,29].

A more recent concern is the increased adoption of food supplements in diets [30]. Augustsson and co-workers [31] demonstrated a considerable variability in the amount of metals in 138 supplements, with discrepancies between the declared versus measured content ranging from 50% to 150%. This included a large spam concentration for Zn^2+^, Cu^2+/+^, Fe^3+/2+^, and Mn^3+/2+^ in supplements, with P95/P5 ratios (P95 percentile to P5 percentile ratio) of 40.000-, 19.000-, 16.000-, and 18.000-fold, respectively. This results in values above the tolerable daily intake (TDI), namely, 10% for copper and 50% for iron, manganese, and zinc, in both normal and underweight women and children (3–6 years old), according to the European Food Safety Authority (EFSA), the US Environmental Protection Agency, the Dutch RIVM, and the National Institute of Public Health and the Environment [31].

Transition metal intoxications at the cellular level may arise from the interaction with proteins in different cellular pathways, which cause changes in enzymatic activities, in protein structure, or even lead to the misplacement of other metals that are important co-factors for different enzymes. Currently, the main therapeutic approach to treating metal intoxication is the use of chelators that can, with limitations, remove metals from the body. The use of natural antioxidants has proven beneficial effects in alleviating the consequences of metal toxicity, such as oxidative stress, particularly in pre-clinical studies. This review provides an overview about the essential transition metals’ (Fe^3+/2+^, Zn^2+^, Cu^2+/+^, and Mn^3+/2+^) metabolism, the toxic consequences derived from metal overload, current treatments, and paths towards novel complementary therapies.

## 2. Essential Transition Metals in Metabolism and Disease

### 2.1. Iron (Fe)

Iron enters the body via dietary sources in two major forms: heme and nonheme iron. The bioavailability of heme-Fe^2+^ from meat, fish, and poultry is around 20–40%, while non-heme-Fe^3+^ present in vegetables, fruits, and cereals has a lower bioavailability of ca. 15% [32]. Iron absorption can be hindered by other metals (calcium and zinc), phytates, polyphenols, and proteins such as conglycinin, present in soybeans [1,33], but is augmented by vitamin C, which enhances the absorption of the nonheme form. Besides its ability to reduce Fe^3+^ for cellular uptake, thereby increasing its bioavailability, vitamin C also stimulates ferritin synthesis and the transferrin (Tf)-dependent Fe^3+^ uptake [34]. Otherwise, Fe^3+^ reduction is mostly performed by ferrireductase duodenal cytochrome b (DCYTB)/six transmembrane epithelial antigen of the prostate (STEAP), allowing Fe^2+^ uptake via the divalent metal transporter 1 (DMT1), while heme-Fe^2+^ is transported by the heme carrier protein 1 (HCP1) into enterocytes (Figure 1A). 

From there, Fe^2+^ is either stored in ferritin, which can accommodate up to 4500 atoms in its ferric form (Fe^3+^), or released into the portal vein via ferroportin-1 (FPN1), which may be repressed by hepcidin (Figure 1A). Iron can also be exported from the intestine via HCP1, followed by Tf binding and subsequent transport in plasma. From the diet, around 1–2 mg/day are absorbed, while roughly the same amount is lost via enterocyte and skin desquamation, hemorrhages, and parasitic infections [35]. The major Fe^3+/2+^ portion is present in erythrocytes (2–2.3 g) that are recycled in the spleen, releasing Fe^3+^ to be bound to Tf (~4 mg). Around 1 g of Fe^3+^ is stored in hepatocyte ferritin, while 150 mg are found in the bone marrow [35]. Iron is essential for hemoglobin synthesis, oxygen transport, biosynthesis of collagen, myelin, neurotransmitters, and several components of the mitochondrial electron transport chain [33].

These essential Fe-dependent metabolic functions appear evident from numerous studies, for example, linking obesity to iron deficiency [36,37]. Interestingly, this linkage cannot be attributed solely to a lower dietary iron uptake due to general malnutrition but rather may be explained by reduced iron absorption due to increased hepcidin levels in obese individuals [38,39]. Increased hepcidin expression appears to be linked to inflammation in adipose tissue, greatly mediated by interleukin 6 (IL-6) [38,40]. Hepcidin levels are also controlled by the gut hormone leptin [41,42], which is commonly dysregulated in obese individuals. This notion is further supported by the observation that serum concentrations of Fe^3+/2+^ and Tf were negatively associated with leptin. This association was found in individuals with a body mass index (BMI) below 30 kg/m^2^ [43].

Conversely, the excess of iron can lead to health problems as well, especially because of the lack of a physiologic excretion route. A consequence of the gradual increase in Fe^3+/2+^ in the body is the aggravation of anemia as a result of lower Fe^3+/2+^ absorption in the gut [44], with its accumulation in the liver and heart, further worsening the course of liver diseases and other chronic conditions such as metabolic and cardiovascular diseases [45]. Higher intake of heme-Fe^2+^ was also associated with a higher risk of type 2 diabetes mellitus in humans [46]. In the brain, excess Fe^3+/2+^ was found in the substantia nigra of patients with Parkinson’s disease [47,48], and its accumulation was associated with tau accumulation and amyloid-β aggregation in patients with Alzheimer disease [49,50].

The most predominant Fe^3+/2+^ overload disorder is hereditary hemochromatosis, with an incidence of 1:220–250 individuals, most commonly observed in populations of northern European origin [51]. Different types of hemochromatosis are known and relate to different disease severity levels. The most common type is linked to mutations in the homeostatic iron regulator (HFE), followed by mutations in hemojuvelin (HJV), hepcidin, transferrin 2 receptor (TfR2), and FPN1 [52]. If such gene mutations negatively affect hepcidin synthesis (e.g., HJV), circulatory iron can rapidly reach high levels (Figure 1B), causing an early onset of disease (first–second decade of life) that impacts different organs such as the heart and the endocrine glands. In the cases where HFE is mutated, a milder late-onset phenotype arises [53]. Transferrin saturation is observed in patients with hemochromatosis, with a later increase in serum ferritin, which indicates iron accumulation in tissues. The most common symptoms include fatigue, malaise, arthralgia, and hepatomegaly. Together with high serum ferritin levels, patients might develop liver fibrosis, which can progress to liver cirrhosis and hepatocellular carcinoma [53]. Thalassemia syndrome also correlates with Fe^3+/2+^ overload due to ineffective erythropoiesis that is caused by mutations in α- or β-globin genes [54].

The typical treatments for Fe^3+/2+^ overload are phlebotomy, whereby blood is taken weekly (around 500 mL), and chelation therapy. The three chelating agents approved by the US Food and Drug Administration (FDA) are deferoxamine (DFO), deferiprone, and deferasirox. Side effects of these chelators are retinopathy and auditory toxicity, neutropenia and agranulocytosis, gastrointestinal issues, and liver and kidney toxicity, respectively [52]. New treatments for Fe^3+/2+^ overload conditions include the development of hepcidin mimetics or agonists that may lead to novel treatments for hereditary hemochromatosis, β-thalassemia, and other diseases of Fe^3+/2+^ excess [55,56]. With proven positive effects in pre-clinical studies, and with a phase I clinical trial in healthy individuals, the ferroportin inhibitor vamifeport (VIT-2763) is currently being studied as a new potential therapy [57,58,59].

### 2.2. Zinc (Zn)

The total amount of zinc (Zn) in an average adult is around 2.5 g (70 kg), with its major portion present in the musculoskeletal system (49.5%), followed by bone (36.7%), skin (4.2%), liver (3.4%), blood (1.5%), and brain (0.6%) [1,60]. In plasma, Zn^2+^ binds predominantly to albumin, α-macroglobulin, and Tf. Intracellularly, its majority is stored in Zn^+2^-binding proteins in the cytosol (Figure 2). 

Zinc absorption in the intestine is highly regulated, and typically 16% to 50% of Zn^2+^ present in the diet is taken up systemically, depending on individual requirements [60]. A meat-based diet promotes Zn^2+^ absorption, while the presence of other metals (calcium and Fe^3+/2+^) or phytates (source of phosphorus in seeds and plants, which form pH-dependent complexes with Zn^2+^) can hinder its absorption [61].

Since there is no body compartment dedicated to Zn^2+^ storage, the metal needs to be constantly replenished from food, thus the balance between absorption and excretion is tightly controlled. Zinc loss occurs mainly via fecal and urinary excretion, menstrual flow in women, and semen in men, as well as loss through hair and nails, and skin desquamation [1,60,61]. Zinc homeostasis is maintained between Zn transporters (ZnT-) and the Zrt-, Irt-like protein family (ZIP), as well as metallothioneins (MTs), whose expression is induced by Zn^+2^ via the metal regulatory transcription factor 1 (MTF-1). At the enterocyte level, ZIP4 transports Zn^2+^ from the intestinal lumen, either to bind cytosolic MT for storage, or to be further transported to vesicles and ER by ZnTs. At the basolateral side, ZnT-1 exports Zn^2+^ into the bloodstream, while ZIP5/ZIP14 may import Zn^2+^ back into the cytosol (Figure 2) [62]. Zinc is a co-factor for over 300 proteins, participating in a multitude of physiological processes in the cell. It is involved in signal transduction for endocrine regulation, systemic growth, response to infection and inflammation, and cytokine production, as well as for gene regulation responsible for synaptic plasticity and neuronal death, mood, and memory regulation [62,63].

Zinc also guides different metabolic functions that regulate obesity and diabetes [64]. It is involved in insulin secretion and subsequent action in peripheral tissues. It also modulates the absorption of long-chain polyunsaturated fatty acids levels through its action in fatty acid absorption in the intestine and subsequent desaturation. Furthermore, Zn^2+^ is important in the assembly and clearance of chylomicrons and lipoproteins [64]. In ob/ob mice, an animal model for obesity, Zn^2+^ concentrations were shown to be reduced in various tissues [65], and intestinal Zn^2+^ absorption was slightly increased [66]. Nonetheless, zinc and its supplementation as therapy in obesity is controversial. Zinc has been shown to increase body fat content and aggravate obesity in genetically obese and dietary-induced obese mice [67]. Moreover, zinc plasma levels were found to be directly correlated with abdominal adiposity and liver fat [68], as well as with the risk of metabolic syndrome in patients, together with copper and iron [18]. Also, studies with zinc supplementation in high-fat diet (HFD)-fed mice have reported different results. Bolatimi and co-workers showed that zinc supplementation in HFD-fed mice did not improve glucose handling, hepatic steatosis, or overall diet induced-liver injury (plasma transaminases) [69], whereas Qi et al. observed that zinc supplementation promoted glucose absorption, reduced lipid deposition, improved HFD-induced liver injury, and regulated energy metabolism [70]. In patients with obesity, Khorsandi et al. [71] demonstrated a beneficial effect on body weight and, more recently, Bashandy et al. [72] reported that zinc nanoparticles reduced body weight, BMI, and leptin concentrations in an obese mouse model through a decrease in inflammation, insulin resistance, and Fe^2+^ cardiac content, along with an increase in cardiac-reduced glutathione (GSH) and Cu^+^/Zn^2+^ superoxide dismutase (SOD1) [72]. Zinc supplementation was also found to have beneficial health effects in reducing the risk of digestive tract cancers and diabetes in adults [73].

In the brain, high-dose Zn^2+^ supplementation (60 ppm in water) in mice has been shown to induce hippocampal Zn^2+^ deficiency. While being mechanistically unclear at present, this caused a deficit of synaptic releasable Zn^2+^, possibly leading to the inhibition of brain-derived neurotrophic factor (BDNF) signaling, resulting in learning and memory deficits [74]. In Alzheimer’s patients, an increase in Zn^2+^ abundance in the brain is associated with the accumulation of the amyloid β-peptide and disease severity (Figure 2B) [75]. In bipolar disorder, in contrast, elevated serum Zn^2+^ levels were observed in clinically stable patients [76]. Thus, with respect to brain damage, both Zn deficiency and excess appear to be detrimental.

Zinc intoxication in humans is mostly due to acute, short-term exposure to Zn^2+^ salts, often by attempted suicide, leading to severe gastrointestinal and pancreatic damage [77,78,79]. Exposure to toxic industrial fumes containing Zn oxide may temporarily impair lung function; contact with Zn^2+^-containing products may provoke skin reactions; and highly concentrated Zn sulfate solutions (20%) may cause ocular damage [79,80]. Chronic and sub-chronic Zn^2+^ exposure is often related to unsupervised and/or long-term supplementation. In the intestine, Zn^2+^ related up-regulation of MTs leads to MT-Cu binding which causes decreased Cu^+^ absorption (Figure 2B). In this scenario, Cu deficiency can be responsible for the development of anemia and neurological problems [81,82]. Zinc intoxication can be treated using Zn chelators such as calcium disodium edetate (CaNa_2_EDTA) or diethylentriamene pentaacetate (DTPA). To directly target the symptoms of Zn^2+^ intoxication, antiemetics, proton pump inhibitors (PPIs) and/or H2-blockers are administered in the case of oral ingestion. A whole-bowel irrigation can also be performed if significant gut burden is observed. Intoxication by inhalation is treated with antipyretics, oral hydration, and nonsteroidal anti-inflammatory drugs. Chronic Zn^2+^ exposure is primarily treated with Cu sulfate and/or Zn chelators in more severe cases [83].

### 2.3. Copper (Cu)

Copper (Cu) is taken up by the diet, with a daily average intake of around 1 mg, and a bioavailability of 65% to 70% [84]. The body Cu^2+/+^ content can be around 100 mg, mainly distributed amongst the liver (≈ 10 mg), muscle (≈ 28 mg), and bone tissue (≈ 46 mg) [85]. Copper is an essential enzymatic co-factor, and approximately 54 Cu-binding proteins have been identified [86], with the most important/studied cuproenzymes being cytochrome c oxidase (CCO), tyrosinase, dopamine-β-hydroxylase, amine oxidase, lysyl oxidase, SOD1, hephaestin, and ceruloplasmin (Cp). These enzymes are fundamental in different metabolic pathways, such as cellular respiration, melanin synthesis, dopamine conversion, oxidoreductase activity, and Fe^3+/2+^ metabolism [87]. The balance between Cu^+^ uptake, distribution, and excretion is well accomplished by the body, crucially involving the intestine and liver. Copper is mainly taken up via nutrition, at the proximal part of the intestine, where it is reduced by a specific family of metalloreductases, the STEAP proteins, and transported by the high-affinity copper transporter 1 (CTR1). Other transporters, like DMT1, or the low-affinity copper transporter 2 (CTR2), have been suggested to have a role in Cu^+^ uptake as well; however, their exact mechanisms remain unclear (Figure 3A) [88]. 

For distribution throughout the body, Cu^+^ is first delivered via ATPase copper transporting alpha (ATP7A) from the enterocytes to the portal vein, where it binds to serum proteins, such as macroglobulin and albumin, or amino acids like histidine (His) (Figure 3A). Afterwards, Cu^+^ enters the liver via CTR1 [89]. Intracellular Cu^+^ binding and transfer amongst Cu-dependent proteins is regulated based on a gradient of increasing Cu-binding affinity [90]. Such “Cu-chaperones” bind Cu^+^ and transport them either to cytosolic enzymes or transporters that deliver Cu^+^ to different enzymes in the cellular organelles [90]. For instance, antioxidant protein 1 (ATOX1) is responsible for the transport of Cu^+^ to ATP7A and ATPase copper-transporting beta (ATP7B) in the trans-Golgi network (TGN). There, apo-Cp can bind up to six Cu^+^ atoms, turning into holo-Cp, that is responsible for the Cu^+^ distribution to other tissues [91,92]. Ceruloplasmin is considered a main copper distributor to other organs in the body, together with serum albumin, α-2-macroglobulin, and His-containing proteins (Figure 3) [93]. Also, ATOX1 is responsible for transferring Cu^+^ to the nucleus or to secretory pathways via ATP7B. The copper chaperone for SOD1 (CCS) distributes Cu^+^ to SOD1 in the cytosol and mitochondria, while a myriad of other different proteins (Cox17, Cox11, Sco1, and Sco2) are responsible for Cu^+^ delivery to CCO into mitochondria (Figure 3).

In a scenario of copper excess, Cu-ATPases (ATP7A and ATP7B) increase Cu^+^ transport to the secretory pathways to be incorporated into cuproenzymes or translocate from the TGN to the vesicular compartment for Cu^+^ excretion [91,94]. Daily, and under physiological conditions, the bile transports around 0.6–6 mg Cu^+^ into the gastrointestinal tract (GIT), with saliva and gastric and pancreatic juices accounting for 0.8 mg, and duodenal secretions accounting for 0.16 mg Cu^+^. Through the fecal route, 0.6–1.6 mg Cu^+^ is eliminated, whereas only 0.05 mg are excreted via the urine [84,95].

Copper excess can lead to various diseases, most prominently Wilson’s disease (WD), a rare autosomal recessive inherited metabolic disorder characterized by the pathological accumulation of Cu^+^ due to mutations in the *ATP7B* gene (Figure 3B) [91,96,97], resulting in hepatic, neurologic, and/or psychiatric symptoms [98]. The treatment of WD involves chelator agents (D-penicillamine and trientine) and zinc salts, aiming at avoidance of Cu^+^ overload and its symptoms [99,100]. In addition to WD, Indian childhood cirrhosis and idiopathic copper toxicosis are also characterized by Cu^+^ excess, which are caused by a synergy of an autosomal-recessive inherited defect in Cu^+^ metabolism and excess dietary Cu^+^ intake [101,102].

Acute effects of excess Cu^2+^ ingestion in humans include GIT symptoms such as nausea or abdominal pain, vomiting, and diarrhea [102]. In more critical situations, the symptoms can progress to hepatic necrosis, neurological diseases, renal failure, and hematological and cardiovascular disorders [103,104,105]. These cases are frequently associated with attempted suicide [106], and the treatment may rely on chelators, such as D-Penicillamine, dimercaprol, ethylenediaminetetraacetic acid (EDTA), and 2,3-dimercaptopropane-sulfonate. Therapy may also include gastric lavage, vasoactive and antiemetic drugs, and hemodialysis [104,106,107,108].

Chronic copper toxicity has also been documented in patients with renal failure receiving dialysis via copper tubing [109], upon exposure to pesticides containing copper [110], and in patients receiving intravenous total parenteral nutrition for long time periods [111,112]. Altered Cu^+^ levels have also been implicated in metabolic syndrome. Both low and high dietary intake have been proposed to increase the risk of developing obesity, referred to as a U-shaped association [113]. In a cross-sectional study, Bulka et al. found a positive correlation between serum Cu^+^ levels and abdominal obesity [114], which is supported by Fan et al. and Övermöhle et al., who showed a strong positive association between Cu^+^ serum levels and obesity in children [115] and adolescents [68,115]. In general, the risk for the development of metabolic syndrome has been attributed to higher serum copper levels [18]. Furthermore, urine copper was demonstrated to positively correlate with lipid accumulation products and the visceral adiposity-, body roundness-, conicity-, body adiposity-, and abdominal volume indices [116].

### 2.4. Manganese (Mn)

The recommended oral intake of manganese (Mn) for adult men is 2.3 mg/day, and for adult women is 1.8 mg/day. The upper tolerable intake in adults is 11 mg/day, with toxicity being observed above 40 mg/day. Manganese can enter the body via absorption in the GIT tract (3% to 5% of ingested Mn^3+/2+^ is further taken up, depending on the individual status), through the lungs after exposure to Mn-rich environments, as happens near smelters, or after dermal contact. Manganese also has the particularity of having 11 known oxidative states, ranging from −3 to +7, with Mn^3+/2+^ as the most physiologically relevant states [117]. In the gut, Mn^2+^ is transported via DMT1, ZIPs (most prominently ZIP8 and ZIP14), and calcium channels. Manganese can also be absorbed as Mn^+3^, bound to Tf, via the transferrin receptor (Figure 4) [118]. 

For the basolateral transport in the intestine, it shares Fe^+2^ and Zn^2+^ transporters such as ferroportin and zinc transporter 10 (Znt10) [119]. Importantly, it is documented that Fe status influences Mn^3+/2+^ metabolism, since they share both intestinal importers and exporters [120]. In the human body, the amount of Mn^3+/2+^ is around 10–20 mg, and its metabolism is highly dependent on the liver, where it accumulates and can be excreted via the bile (3.6 mg/day) [121]. In blood, its concentration can range from 1.6 to 62.5 μg/L, depending on age, ethnicity, and gender [122]. Manganese accumulates non-uniformly in the brain (5.32 to 14.03 μg/g wet weight), namely in the striatum, globus pallidus, substantia nigra, and hypothalamic nuclei [117]. This essential trace element is involved in the synthesis and activation of various enzymes, regulation of glucose and lipid metabolism and acts as a cofactor for enzymes like arginase, glutamine synthase, and manganese superoxide dismutase (MnSOD) (Figure 4) [2,123]. The metalloenzyme MnSOD is localized in the mitochondria and is one of the most important antioxidant components in the cell. This enzyme catalyzes the dismutation of superoxide in hydrogen peroxide, which is further degraded by catalase (CAT) to form water and oxygen [2].

Manganese imbalance, particularly Mn^2+^ excess (hypermanganesemia), has been associated with neurological and behavioral defects as well as diseases such as Parkinson’s disease (PD) and manganism [124,125,126,127,128], and reproductive [129] and respiratory problems in both humans and animals [130]. Manganese dyshomeostasis has also recently been implicated in metabolic syndrome [2]. Higher dietary Mn^3+/2+^ intake has been shown to increase the risk of developing metabolic syndrome [131]; elevated blood levels of Mn^2+^ were shown to correlate with increased visceral adipose tissue [132]; and urinary manganese was positively associated with metabolic syndrome in Asian women [133].

On one hand, as an important co-factor for MnSOD, its deficiency can lead to mitochondrial dysfunction due to a decreased capacity for reactive oxygen species (ROS) scavenging. On the other hand, being a highly reactive element, in excess, it can cause an increase in ROS production, ultimately damaging mitochondria (Figure 4B) [2,127]. Manganese intoxication can occur in individuals with chronic liver disease due to the failure of its hepatic clearance, and upon prolonged total parenteral nutrition, where excessive Mn^2+^ amounts bypass the hepatic filter and enter the bloodstream [134]. Furthermore, occupational exposure to excess Mn^7+/3−^ poses a risk for factory workers, miners, and welders [135,136]. Other human activities can expose individuals to high Mn^7+/3−^, including the use of fungicides, medical imaging contrast agents and water purification agents, the combustion of gasoline, and Mn-containing emissions from contaminated soil, dust, and plants near roadways. Treatment for Mn^3+/2+^ intoxication is based on chelation therapy with EDTA and, in some cases, administration of Fe^3+/2+^ can be used in combination with chelators [119,137,138]. In Chinese patients, para-amino salicylic acid (PAS) proved to be a promising treatment for severe Mn intoxication [139].

## 3. Essential Transition Metals and Oxidative Stress

Metal toxicity has been linked to oxidative stress [140,141]. In cells, the generation of ROS is a physiologic mechanism derived from the utilization of oxygen by different metabolic reactions [142]. In response, the antioxidant enzymes control the harmful effects of ROS (Figure 5A), which may also lead to DNA damage, lipid peroxidation, mitochondrial dysfunction, and cell death (Figure 5B) [142]. 

The labile Fe^2+^ pool can be present in different cellular compartments, such as cytosol, mitochondria, and lysosomes, and prompts the participation of the metal in reactions of oxidation and reduction, which can catalyze the formation of hydroxyl radicals (OH) from hydrogen peroxide via the Fenton and Haber/Weiss reactions (Figure 5C) [143,144]. Iron overload was shown to cause oxidative stress in skeletal muscles, delayed muscle regeneration, decreased expression of myoblast differentiation markers, and decreased phosphorylation of MAPK signaling pathways in a mouse model of cardiotoxin-induced muscle regeneration [145]. An increase in ROS, as well as decreased insulin signaling, was observed upon iron treatment of mouse hepatocytes [146]. Additionally, oxidative stress related to iron toxicity was implicated in neurodegeneration [147], kidney injury [148], and delayed spinal cord regeneration [149].

In mitochondria, biogenesis of Fe-sulfur cluster proteins, which are fundamental for their function, are highly sensitive to an increase in ROS and Fe^3+/2+^ levels [150,151]. To this end, it is not unforeseen that Fe-induced oxidative stress led to mitochondrial dysfunction in cardiac tissues. Chan and co-workers [152] observed decreased mitochondrial function and increased oxidative stress in embryonic heart H9C2 cells, plus opening of the mitochondrial permeability transition pore in ventricular myocytes from mice treated with Fe^3+^/8-hydroxyquinoline. Furthermore, Gordan et al. described arrhythmias in ex-vivo mouse hearts, also upon treatment with Fe^3+^/8-hydroxyquinoline [153]. A reduced respiratory capacity of hepatic mitochondria, together with an increase in ROS was observed by Volani and co-workers in mice fed with an Fe-enriched diet for two weeks [154].

For Cu^2+/+^, in the case of an intracellular elevation of the metal, GSH and MTs bind it with high affinity [155,156], largely avoiding the involvement of non-bound Cu^+^ in Fenton-based reactions. For a long time, oxidative stress was considered as the main mechanism of cellular damage in Cu-overload toxicity, as happens in WD. However, decreased activity of antioxidant enzymes, with increased lipid peroxidation and DNA damage were only observed at later stages of the disease, in both WD patients and animal models [157,158]. Moreover, studies in yeast showed that free Cu^+^ concentration is less than 10^−18^ M, which corresponds to less than one free Cu^+^ atom per cell, rendering the cellular free Cu^+^ pool practically nonexistent [159]. Therefore, even in copper stress conditions, the cellular response capacity towards Cu-related oxidative damage has proven efficient, at least in an early disease state [157]. 

Mitochondria are indeed a preferential target for Cu^+^ toxicity in cells. New evidence shows that specific proteins in the mitochondria could be targets for Cu^+^, namely thiol-rich [158,160,161,162] or lipoylated proteins [163], resulting in proteotoxic stress and cell death (Figure 5D). In a study by Borchard and co-workers, it was observed that mitochondria isolated from brain tissue were more sensitive to Cu^+^ challenges in comparison to those from heart, kidney, and liver tissue. Mitochondria structural alterations were present upon in vitro treatments, with a Cu/GSH ratio of 1:10, i.e., at reducing conditions, thereby ruling out a Fenton chemistry-based mechanism of destruction. Interestingly, a significant emergence of ROS was only detected upon treatment with a Cu/GSH ratio of 5:10, which would hardly occur in vivo [160]. Such results clearly argue for a thiol/protein-directed attack of Cu^+^ as the toxic mechanism, and not for an undirected overwhelming oxidative stress via Cu-induced Fenton chemistry. Furthermore, other mechanisms for Cu-induced toxicity were described, such as the interference with cellular signaling pathways, like the mitogen-activated protein kinases (MAPKs) pathway [164], or the induction of apoptosis via the mitochondria apoptotic pathway [165,166].

In the case of Zn^2+^, although it is generally considered a redox-inert metal, it serves as a co-factor for several enzymes that participate in redox reactions, namely SOD1. It has a complex and important role in the oxidative stress balance in cells, both as an antioxidant and as a prooxidant [167]. In the form of Zn^2+^, it can bind to and inhibit mitochondria complex I, III, and IV, which can result in decreased ATP production, increased mitochondrial membrane permeability transition (MPT), and ROS generation (Figure 5D). Moreover, Zn^2+^ overload can impact glycolysis by inhibition of glyceraldehyde 3-phosphate dehydrogenase (GAPDH), pyruvate kinase, and phosphofructokinase [167]. Due to allosteric similarities, zinc can also compete for copper and iron binding sites, causing its misplacement in a scenario of Zn^2+^ overload (Figure 5D). Overall, excessive Zn^2+^ directly affects several signaling pathways that ultimately lead to disruption of cellular homeostasis [167]. Chen et al. reported that Zn sulfate elicited oxidative stress, decreased mitochondrial membrane potential and induced the activation of extracellular signal-regulated kinases 1 and 2 (ERK1/2) phosphorylation, lipid peroxidation, and DNA oxidation in human neuroblastoma cells [168]. Slepchenko and co-workers also observed an increase in ROS upon zinc treatment in HeLa cells subjected to hypoxia. In this study, the inhibition of the nicotinamide adenine dinucleotide phosphate (NADPH) oxidase activity significantly decreased Zn-induced ROS, which led the authors to hypothesize that intracellular activation of this enzyme by zinc triggers mitochondrial ROS production [169]. In agreement, Noh et al. showed that Zn^2+^ overload induced NADPH oxidase activation in a protein kinase C (PKC)-dependent manner, possibly contributing to the increased generation of ROS in mouse cortical cultures [170]. In a study by Pan et al., zinc was also shown to induce mitochondrial ROS, as well as decreased mitochondrial membrane potential in hypoxia-induced astrocytes [171].

Manganese, as an essential redox-active trace metal, has also been implicated in inducing oxidative stress in cellular systems. A time-dependent increase in intracellular ROS/reactive nitrogen species (RNS), decreased GSH content, and impaired mitochondria function was reported by Neely et al. in human-induced pluripotent stem cell-derived postmitotic mesencephalic dopamine neurons after treatment with MnCl_2_ [172]. Nuclear localization and subsequent binding of nuclear factor erythroid 2 (Nrf2) to the antioxidant-responsive element leading to heme oxygenase-1 (HO-1) was observed by Li and co-workers in rat catecholaminergic cells upon exposure to 300 μM MnCl_2_ (Figure 5F) [173]. In a study by Tan et al., increasing doses of MnSO_4_ led to a decrease in SOD, glutathione peroxidase (GPx), and CAT activities, while the levels of malondialdehyde (MDA) were upregulated in rat adrenal pheochromocytoma-derived cells. Moreover, cell apoptosis was significantly increased, as shown by the significant decrease in B-cell lymphoma 2 (Bcl-2) and caspase-3 mRNA levels, while Bcl-2-associated X protein (Bax) mRNA levels increased [174]. In addition, Liu and co-workers have also shown that Mn^2+^ induced a significant increase in H_2_O_2_ production in the mitochondria of rat microglia cells through suppression of complex II (Figure 5D) [175].

## 4. Antioxidant Therapies for Essential Transition Metals Toxicity

### 4.1. Iron (Fe)

Iron is a redox-active essential transition metal that is indispensable for several biological processes, but can also induce oxidative stress when in excess [176]. Rhee et al. showed that the DMT1 inhibitor and antioxidant ebselen (Figure 6) prevented intracellular Fe^2+^ uptake and decreased ROS production in human induced pluripotent stem cell-derived cardiomyocytes (iPSC-CMs) subjected to Fe^3+/2+^ overload. In humans, iron overload induced arrhythmia and contractile dysfunction, similarly to the effects observed in the cardiomyocytes [177]. Natural flavonoids like quercetin (and some derivates), catechin, and rutin (Figure 6) had a positive antioxidant effect on iron excess in human red blood cells. In this study by Cherrak et al., Fe^3+^ and Zn^2+^ had an enormous prooxidant effect (37% and 33% induced hemolysis, respectively), while treatment with the flavonoids had a pronounced antioxidant activity against iron, as measured by levels of hemolysate-reduced GSH and MDA, as well as CAT activity [178]. Flavonoids have been repeatedly shown to protect against Fe toxicity, either by direct metal chelation or by scavenging of oxidant species [179,180]. In pre-clinical models, depending on the compound, different effects were described. Wang and co-workers observed that treatment with myricetin reduced iron content and inhibited transferrin receptor 1 (TfR1) in human neuroblastoma cells, and significantly reversed scopolamine-induced cognitive deficits in a mouse model of Alzheimer’s disease [180]. In two different studies, myricetin (Figure 6), present in tomatoes, oranges, nuts, berries, tea and red wine, significantly inhibited hepcidin expression in vitro and in vivo [181], thereby possibly preventing iron accumulation in the brain [182]. Two different dithiolethiones (D3T and ACDT), present in cruciferous vegetables, exhibited antioxidant activity by activating Nrf2 transcription factor and upregulating GSH levels, which protected human glioblastoma cells (U-87) from Fe-induced toxicity. Furthermore, Kulkarni and co-workers showed that D3T and ACDT could upregulate the expression of Nrf2-mediated iron storage protein ferritin, resulting in a reduced total labile Fe^2+^ pool, therefore preventing ferroptosis-induced cell death by erastin [183]. Molinari et al. reported the combined action of lipoic acid (Figure 6), present in foods such as red meat, carrots, beets, spinach, broccoli and potatoes, and vitamin D to decrease intracellular iron content and ROS production in primary mouse astrocytes. Furthermore, p53 activity, amyloid precursor protein and SOD1 content was significantly reduced in comparison to the Fe-treated control [147].

The amino acid taurine (Figure 6), enriched in shellfish, turkey and chicken, has an important role in aging, cardiovascular health, neuroprotection, and cellular function, as shown in murine models and humans in a detailed overview by Santulli et al. [197]. The benefits of taurine are correlated with its capacity to act as an osmolyte, regulating cell volume and maintaining cell integrity; its antioxidant properties; and its role in calcium signaling and neurotransmission, as well as in bile acid metabolism [197]. Zhang et al. also described the hepatoprotective properties of taurine in an iron-overload murine model [198]. Quercetin (Figure 6), enriched in capers, is one of the most reported antioxidant compounds for Fe-overload treatment, with proven positive effects in iron depletion and oxidative stress in rats, mice, and human carcinoma cell lines [199,200,201]. In thalassemia patients under DFO therapy, Sajadi et al. reported that 500 mg/day of quercetin for 12 weeks reduced high-sensitivity C-reactive protein, Fe content, ferritin and Tf saturation, and increased Tf levels in serum, in comparison to non-treated controls [200]. Moreover, in thalassemia patients at risk of suffering from iron toxicity due to blood transfusions, treatment with a combination of DFO and silymarin (Figure 6, silibinin) from milk thistle for 9 months significantly decreased serum ferritin levels, serum Fe^3+/2+^, and total Fe-binding capacity, in comparison with the placebo group (DFO-alone). Serum hepcidin and soluble Tf were also significantly decreased in the silymarin-treated group, with improvement in overall liver function, in comparison to placebo [202].

### 4.2. Zinc (Zn)

Zinc plays an important role in human health, and it has anti-oxidant and anti-inflammatory properties [203]. In contrast to other essential transition metals, such as Fe^3+/2+^ and Cu^2+/+^, few conditions are linked to Zn^2+^ overload, whether the consequences are related to oxidative stress or not. Nonetheless, some studies have reported the potential of antioxidant therapies to ameliorate Zn-related toxicity. Deore and co-workers showed that administration of α-lipoic acid (15 mg/kg bw) for 15 days improved the levels of SOD, CAT, GSH and GPx, and decreased ROS in spleen and brain tissue of rats treated with zinc oxide nanoparticles (ZnONP), 100 mg/kg bw, for 28 days. Also, the augmented levels of tumor necrosis factor alpha (TNF-α) and interleukins (IL-1β, IL-4, and IL-6) were decreased in rat brain and spleen tissue upon α-lipoic acid treatment [204]. An improvement of the cerebellum structure, together with reduced oxidative stress, e.g., MDA, GPx, and nitric oxide (NO) levels, autophagy (caspase 3, p53) and inflammatory response (IL-1, IL-6, TNF-α), was observed by Amer et al. after pre-treatment with curcumin (200 mg/kg diet), enriched in turmeric root, in rats exposed to ZnONP (5.6 mg/kg bw) for 28 days. However, no differences in brain Zn^2+^ levels were observed upon pre-treatment [205]. Other studies also reported reduced lipid peroxidation, improved oxidative defense systems (SOD and GPx), and reduced apoptosis in rats pre-treated with curcumin (200 mg/kg in the food) and afterwards exposed to ZnONP [206,207]. The use of ZnONP in different applications such as cosmetics, electronics, and in the chemical and medical industry has raised concerns for its potential toxicity for various organisms, from algae and fish to humans. Particularly in mammalian cells, several studies have found that ROS play a key role in ZnONP toxicity [208], which further supports the use of antioxidants as a potential remedy.

### 4.3. Copper (Cu)

In similarity to Fe^3+/2+^, Cu^2+/+^ is also a redox-active essential transition metal that participates in different cellular processes, mostly mitochondrial ROS scavenging and oxidative phosphorylation, as well as different nuclear signaling pathways. Nevertheless, elevated Cu^2+/+^ levels can be toxic and lead to proteotoxic stress and ultimately cell death. Several natural compounds have been investigated for their antioxidant properties in the context of Cu^2+/+^ toxicity. A combination of bioactive antioxidant compounds (resveratrol, ferulic acid, phloretin, and tetrahydrocurcuminoids) (Figure 6), improved cellular viability, increased proliferation, and decreased total ROS emergence in human oral fibroblasts treated with CuCl_2_ [209]. Tamagno et al. further showed that the use of an antioxidant-rich pitaya fruit extract ameliorated Cu-induced toxicity in organisms like *Caenorhabditis elegans* by enhancing the antioxidant system [210]. Treatment with this extract decreased the activity of acetylcholinesterase (AChE), lipid peroxidation, and the levels of SOD and CAT that were elevated upon Cu exposure [210]. A concomitant treatment of the same pitaya extract with 0.7 mg/L Cu^2+^ for four days in zebrafish decreased glutathione S-transferase (GST) and CAT activity in brain tissue, while SOD activity was increased. In gut tissue, GST activity was increased upon treatment, while CAT activity was also decreased, in comparison with zebrafish Cu-treated controls [211]. The activity of AChE and δ-aminolevulinate dehydratase was increased in the brain, while the cortisol levels in the whole body were decreased upon co-treatment with the extract [211]. Moreover, Azeez and co-workers reported that the treatment with vitamin E (Figure 6), in the presence of Cu-sulphate, improved body weight, GPx and GST activity, while decreasing H_2_O_2_ and MDA levels in the liver of *Clarias gariepinus* (African catfish), in comparison with Cu-treated controls [212]. A 21-day treatment with silymarin encapsulated in a liposomal formulation (0.5 mg/kg bw, twice daily) improved liver dysfunction and neurobehavioral abnormalities associated with Cu-toxicity in rats given Cu-sulphate (200 mg/kg bw daily) for 90 days [213].

In a widely used rat model for Cu-overload in LEC rats, therapeutic antioxidant strategies to improve liver damaged were reviewed by Zischka et al. [157]. From the collected evidence, linolenic and linoleic acid supplemented in the diet for 10 weeks reduced the incidence of hepatitis and delayed its onset for one month. However, no effect was observed with regards to oxidative stress [214]. In a study by Yamamoto et al., treatment with lipoic acid dose-dependently reduced liver damage and increased GPx, glutathione reductase (GR), and SOD1 activity, while decreasing lipid peroxidation. A decrease in Cu^+^ and Fe^2+^ content was also observed, but only in mitochondria [215]. Kitamura and co-workers showed that N-acetylcysteine (NAC) (Figure 6) had a positive effect in LEC rats by decreasing the Cu/Fe ratio, as well as liver and kidney damage. However, the authors hypothesized that this effect is most likely linked to metal chelation rather than ROS scavenging [216]. Overall, most antioxidant therapies had only a mild effect in vivo, inducing some delay in the onset of hepatitis, but not reverting liver damage or oxidative stress [157].

### 4.4. Manganese (Mn)

Manganese can induce oxidative stress in cells via H_2_O_2_ production and disruption of different cellular functions such as mitochondria respiration and dopamine oxidation [2,123]. This effect is particularly described for the brain, with Mn^2+^ being associated with neurological problems that are currently attributed mostly to oxidative damage but also to the impairment of several neurotransmitter systems, especially dopaminergic, but also cholinergic and GABAergic [217]. To this end, potential antioxidant therapies have been explored in the context of Mn toxicity.

Chtourou et al. observed the beneficial effects of the flavonoid silymarin against Mn-induced, radical-mediated cell death in murine neuroblastoma cells [218]. Co-treatment with silymarin for 24 h improved cellular viability, reduced cellular H_2_O_2_ levels, and improved MnSOD, SOD1, CAT, and GPx enzymatic activities [218]. A methanolic extract of Acaí (most probably due to its anthocyanins) was reported by da Silva Santos et al. to reduce neurotoxicity in rat primary astrocytes [219] by restoring the GSH/GSSG ratio and net glutamate uptake, and protecting the cell membranes from lipid peroxidation and Nrf2 activation by Mn-exposure. However, higher concentrations of the extract exacerbate the negative effects of Mn^2+^, further pointing to the importance of dosing [219]. Stephenson and co-workers showed that pre-treatment with NAC or GSH reduces the DNA damage induced dose-dependently by Mn^2+^ in human neuroblastoma cells [220]. The natural flavonoid quercetin prevented Mn-induced oxidative stress in the hypothalamus, cerebrum, and cerebellum in the brains of rats, which was characterized by increased H_2_O_2_ and MDA levels, and decreased antioxidant enzyme activity (measured SOD and CAT), with the animals presenting pronounced locomotor impairment and increased AChE activity [221]. Moreover, Bahar et al. observed that pre-treatment with quercetin improved cellular viability and decreased ROS production and MDA levels, while increasing SOD, CAT, and GSH activity in human neuroblastoma cells treated with Mn^2+^ [222]. Furthermore, an improvement in mitochondrial function, and a decrease in apoptosis was observed in the cells. A reduction in inflammatory factors was also observed in both cells and rats treated with quercetin. Particularly in the animals, TNF-α, IL-1β, IL-6, and iNOS protein expression in the brain was reduced in comparison to Mn-treated controls. The mRNA levels of NF-kB and inducible nitric oxide synthase (iNOS) were reduced, while *HO-1* and *Nrf2* were increased in brain tissue, upon treatment. Lastly, quercetin treatment reduced the apoptotic marker expression in the rats, as observed by the decreased protein expression of Bax, cytochrome C, caspase 3, and poly(ADP-ribose)-polymerase 1 (PARP-1) in the brain [222]. The neurohepatic protective effects of the carotenoid lycopene were studied by Lebda et al. in rats treated daily with Mn^2+^ for 4 weeks, after 20-days of lycopene supplementation. No significant changes were observed in Mn^2+^ content in the serum, liver, or brain tissue. But ALT, AST, AChE, and glucose levels were decreased in the serum of pretreated animals in comparison to Mn-treated controls [223]. In the liver and brain, lipid peroxidation was decreased, while GSH, GST and CAT levels were increased as consequence of lycopene pretreatment. The amount of AChE in the brain was increased with Mn^2+^ exposure, and further decreased with the therapeutic intervention [223]. Cordova and co-workers explored the potential positive effects of the hydrophilic vitamin E analog Trolox (Figure 6) on the central nervous system of rodents and cell cultures exposed to Mn^2+^. A concomitant addition of Trolox and Mn^2+^ to rat pups (8 days old) decreased caspase activity and oxidative stress (by F2-isoprostrane quantification) in the striatum, while reversing the motor coordination deficits observed in the animals treated with Mn^2+^ [224]. The positive effects of Trolox and NAC were also observed by Marreilha et al. in immortalized rat brain microvessel endothelial cells exposed to Mn^2+^. Co-treatment with the antioxidants for 24 h improved Mn-induced loss of cell viability and reduced the content of thiol-rich proteins such as GSH [225]. Milatovic and co-workers also reported the neuroprotective effect of Trolox pre-treatment in rats (primary cultures and in vivo treatment) against oxidative damage and ATP depletion caused by the presence of Mn^2+^ [226]. In humans, treatment for three and a half months with sodium para-aminosalicylic acid, 6 g/day, intravenously (4 days treatment, 3 days pause) significantly improved neurological issues such as tremor, muscular tension, difficulties with writing and speaking, and gait and grip capacity, amongst others, derived from chronic exposure to Mn^2+^ [227].

Table 1 summarizes the most relevant antioxidant compounds reported to ameliorate the oxidative damage caused by iron, copper, and manganese and their respective positive effects in pre-clinical and clinical studies, as described in the text.

## 5. Conclusions

Essential transition metals play key roles in cellular metabolism. As enzymatic co-factors, they interfere with lipid and glucose metabolism, mitochondrial function, transcription, protein synthesis, and cellular renewal processes, among others. However, excess caused by genetic mutations, environmental exposure, or over-supplementation can be detrimental and even fatal if untreated. Apart from traditional chelation therapies, the use of antioxidants as a treatment strategy for metal toxicity is supported by research that demonstrated its efficacy in alleviating oxidative damage. Nonetheless, such investigation was mostly done in pre-clinical models, with studies in humans showing the potential of antioxidant therapies almost nonexistent. The reason could be related to the high molecular complexity of metal-induced oxidative stress and the knowledge still lacking in terms of how transition metals, such as manganese, act. Moreover, most of the information available report beneficial effects of antioxidants before the metal insult, or as a co-treatment. The use of antioxidants as a treatment in the case of metal overload and toxicity needs to be thoroughly addressed in future studies.

A potentially promising approach could be the combination of antioxidants with chelating agents, addressing both metal overload and oxidative stress. Clearly, with the current rise of uncontrolled and unsupervised supplementation, as well as the increasing evidence of toxicity by these metals, more efficient therapeutic countermeasures need to be developed.

## Figures and Tables

**Figure 1 ijms-25-07880-f001:**
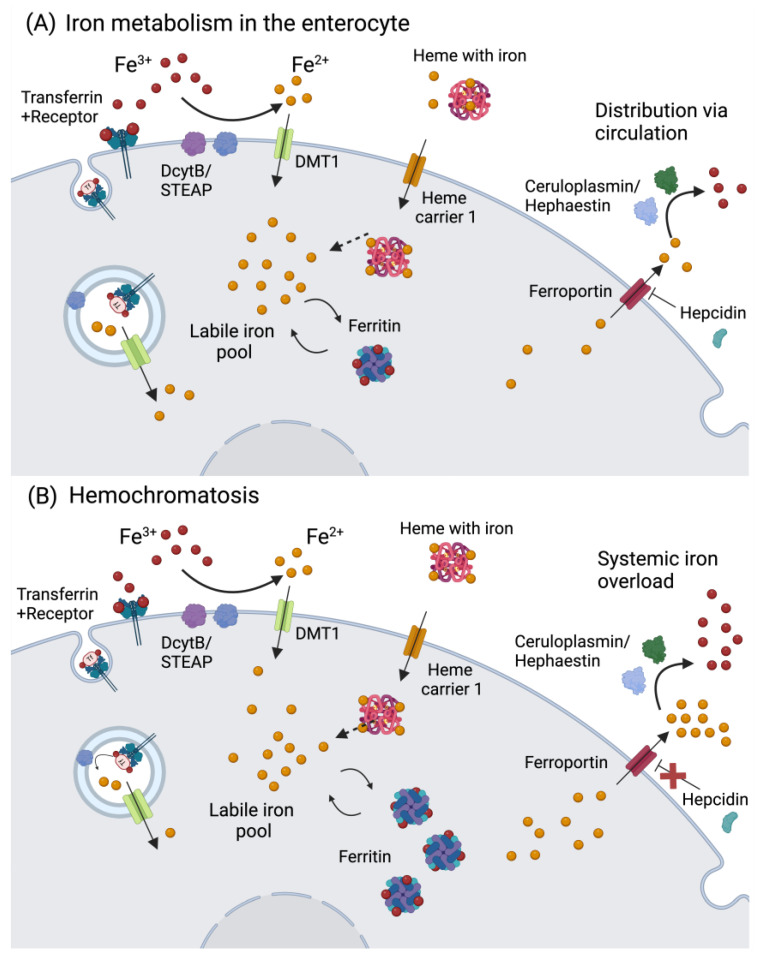
Iron metabolism in the enterocyte and hemochromatosis. (**A**) Iron is absorbed as Fe^2+^ through reduction by DcytB/STEAP, and transported via DMT1. In the case iron is bound to heme, it can be transported via heme carrier 1. Another route is via receptor-mediated endocytosis bound to transferrin. Afterwards, iron can be stored in the form of ferritin, used for the biosynthesis of Fe-S clusters, or integrate the cellular labile iron pool. The iron in the labile iron pool can bind to ferritin, and vice-versa. If iron is not stored or used, it will be exported to the circulation via ferroportin, and oxidized to ferric iron by hephaestin (in the intestine) or ceruloplasmin. Iron export via ferroportin is regulated by hepcidin, which is produced by the liver. (**B**) In hemochromatosis, hepcidin expression is low, thus ferroportin activity is left unregulated, resulting in an increase in iron efflux in the circulation. Abbreviations: DcytB: ferrireductase duodenal cytochrome b; DMT1: divalent metal transporter 1; Fe: iron; STEAP: six-transmembrane epithelial antigen of the prostate. Created in BioRender.com (accessed on 19 May 2024).

**Figure 2 ijms-25-07880-f002:**
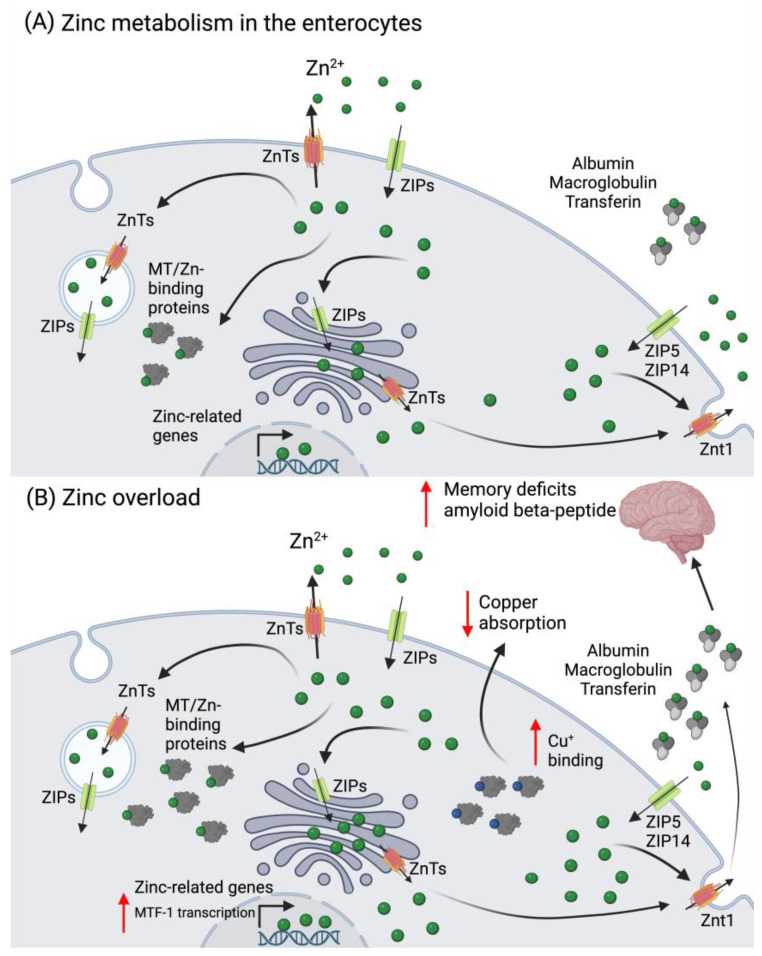
Zinc metabolism in the enterocyte and its overload. (**A**) Zinc is absorbed as Zn^2+^ via the ZIPs (mainly ZIP4) and can be excreted into the intestinal lumen by the ZnTs. In the enterocytes, Zn^2+^ can be found as cytoplasmic free zinc (which can bind to low molecular weight ligants), sequestered into vesicles, or bound to Zn-binding proteins, such as MT. Zinc can also be transported to the Golgi apparatus or to the nuclei, where it induces the expression of several zinc-related genes. In circulation, Zn^2+^ binds to albumin, macroglobulin, or transferrin. (**B**) In a scenario of zinc overload, there is an increase in the transcription of MTF-1, which leads to an increased MT expression to buffer the cytosolic Zn^2+^. The overexpression of MT leads to Cu^+^ binding and decreased absorption. The increase in zinc in circulation can be detrimental for brain function, leading to the development of memory deficits and increased deposition of the amyloid beta peptide, which is linked to different neurodegenerative conditions. Abbreviations: MT: metallothioneins; MTF-1: metal regulatory transcription factor 1; ZIP: Zrt-, Irt-like protein family; Zn: zinc; ZnT: zinc transporter. Created in BioRender.com.

**Figure 3 ijms-25-07880-f003:**
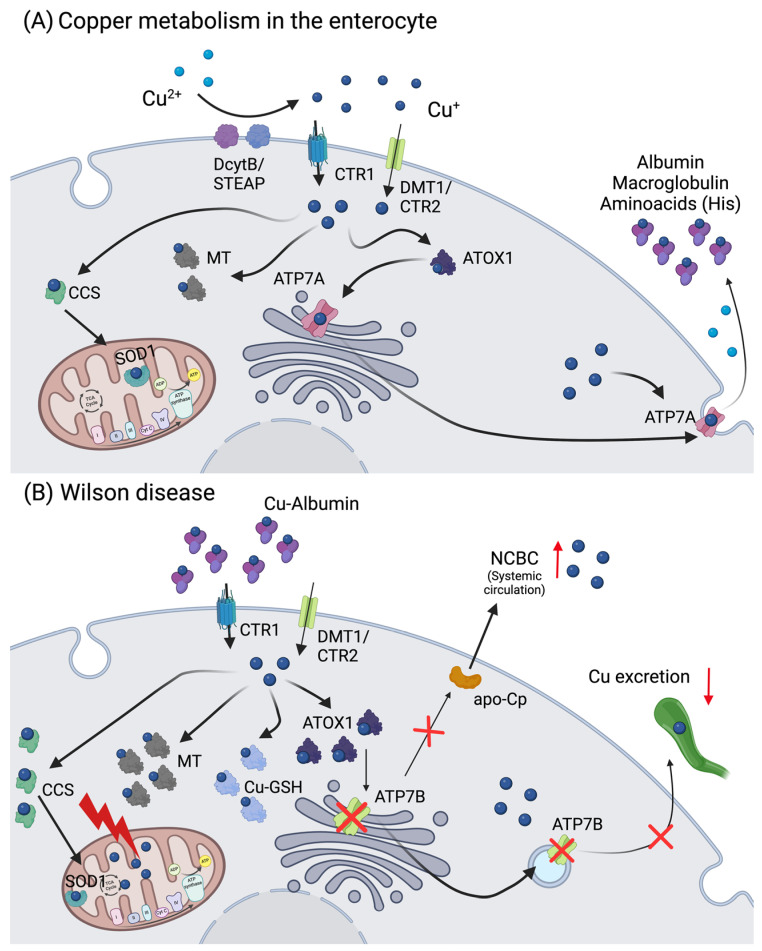
Copper metabolism in the enterocytes and Wilson disease. (**A**) Copper is reduced to Cu^+^ by STEAP or DcytB, and transported mainly via CTR1, but also by DMT1/CTR2. Thereupon, copper can be stored in the cell in the form of Cu-MT, bound to ATOX1, or transported to SOD1 in the mitochondria via CCS. In the case copper is not stored or used, it will be exported via ATP7A, oxidized to Cu^+^ (in the presence of oxygen), and bound to albumin, macroglobulin, or cysteine-rich amino acids such as histidine (His), in circulation. (**B**) In Wilson disease, in one hand, the ATP7B mutation results in reduced Cu incorporation into ceruloplasmin (Cp), increasing the amount of non-ceruloplasmin bound Cu (NCBC) in circulation. On the other hand, copper sequestration and excretion into the bile by ATP7B is strongly inhibited, leading to its accumulation in the hepatocytes. As consequence, MT and GSH levels increase and Cu accumulates in the mitochondria, leading to mitochondrial dysfunction. Abbreviations: ATOX1: antioxidant copper chaperone 1; ATP7A: ATPase copper-transporting alpha; ATP7B: ATPase copper-transporting beta; CCS: copper chaperone for superoxide dismutase 1; apo-Cp: ceruloplasmin; CTR1/2: high-affinity copper transporter 1/2; Cu: copper; DcytB: ferrireductase duodenal cytochrome b; DMT1: divalent metal transporter 1; GSH: glutathione; His: histidine; MTs: metallothioneins; NCBC: non-ceruloplasmin-bound copper; SOD1: superoxide dismutase 1; STEAP: six transmembrane epithelial antigen of the prostate. Created with BioRender.com.

**Figure 4 ijms-25-07880-f004:**
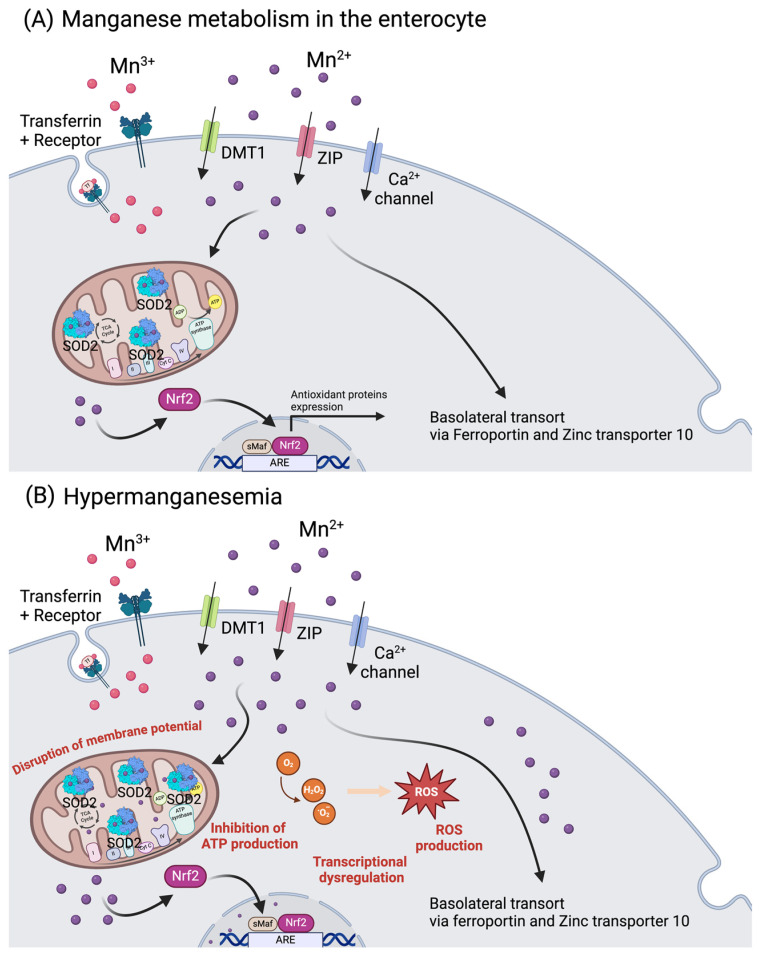
Manganese metabolism and hypermanganesemia. (**A**) Manganese can be taken up as Mn^2+^ via DMT1, ZIP, or Ca^2+^ channels, or as Mn^3+^ by binding to transferrin. In mitochondria, Mn^2+^ works as a cofactor for MnSOD (SOD2). Manganese (Mn^2+^) can also influence the expression of genes (i.e., antioxidant proteins) by binding to nuclear transcription factors. Basolateral manganese release occurs mainly via ferroportin and zinc transporter 10. (**B**) In excess, as happens in hypermanganesemia, manganese accumulates in the mitochondria, leading to inhibition of ATP production and disruption of the membrane potential, and in the nucleus, where it dysregulates gene transcription. Both situations lead to an increase in ROS production. Abbreviations: ARE: antioxidant response elements; Ca: calcium; DMT1: divalent metal transporter 1; Nrf2: nuclear factor erythroid 2; ROS: reactive oxygen species; sMaf: small MAF; SOD2: superoxide dismutase 2; ZIP: Zrt-, Irt-like protein family. Created with BioRender.com.

**Figure 5 ijms-25-07880-f005:**
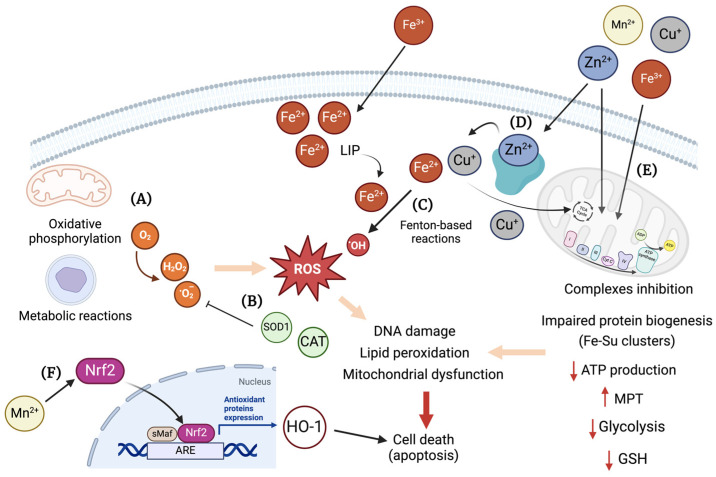
Overview of cellular mechanisms related to toxicity by essential transition metals. (**A**) Different metabolic pathways in the cell use O_2_, e.g., oxidative phosphorylation in the mitochondria. The generation of ROS, such as H_2_O_2_ and •$O_{2}^{-}$, are mostly kept under control by the internal antioxidant system, e.g., SOD1, catalase, etc. (**B**) However, in a scenario of decreased antioxidant cellular capacity, an increase in ROS can elicit damage to proteins, lipids, and DNA, leading to cell death, e.g., via apoptosis. (**C**) Also, through Fenton-based reactions involving transition metals, such as Fe^2+^, •OH is formed, potentially leading to cellular damage. (**D**) Zinc, despite being a redox-inert metal, can cause mitochondrial dysfunction by interfering with the oxidative phosphorylation process, biogenesis of Fe-S clusters, and glucose metabolism. In excess, zinc can displace other metals, like Cu^+^ and Fe^2+^, from antioxidant enzymes, thus causing cellular metal imbalance. (**E**) Manganese, with Cu^+^ and Fe^2+^, is a redox-active metal that targets mitochondria and causes inhibition of the complexes’ activity leading to a decrease ATP production and the increase in mitochondria permeability transition pore (MPT). Furthermore, a decreased glycolytic activity and GSH content can occur as consequence of metal overload. (**F**) Mn^2+^ can also act in cellular signaling by activating Nrf2, which induces the expression of HO-1 through binding to ARE, potentially leading to cell death. ARE: antioxidant response elements, CAT: catalase, Cu: copper, Fe: iron, HO-1: heme oxygenase 1, LIP: labile iron pool, Mn: manganese; MPT: mitochondria permeability transition pore; Nrf2: nuclear factor erythroid 2, ROS: reactive oxygen species, sMaf: small MAF; SOD1: Cu/Zn superoxide dismutase, Zn: zinc. Created with BioRender.com.

**Figure 6 ijms-25-07880-f006:**
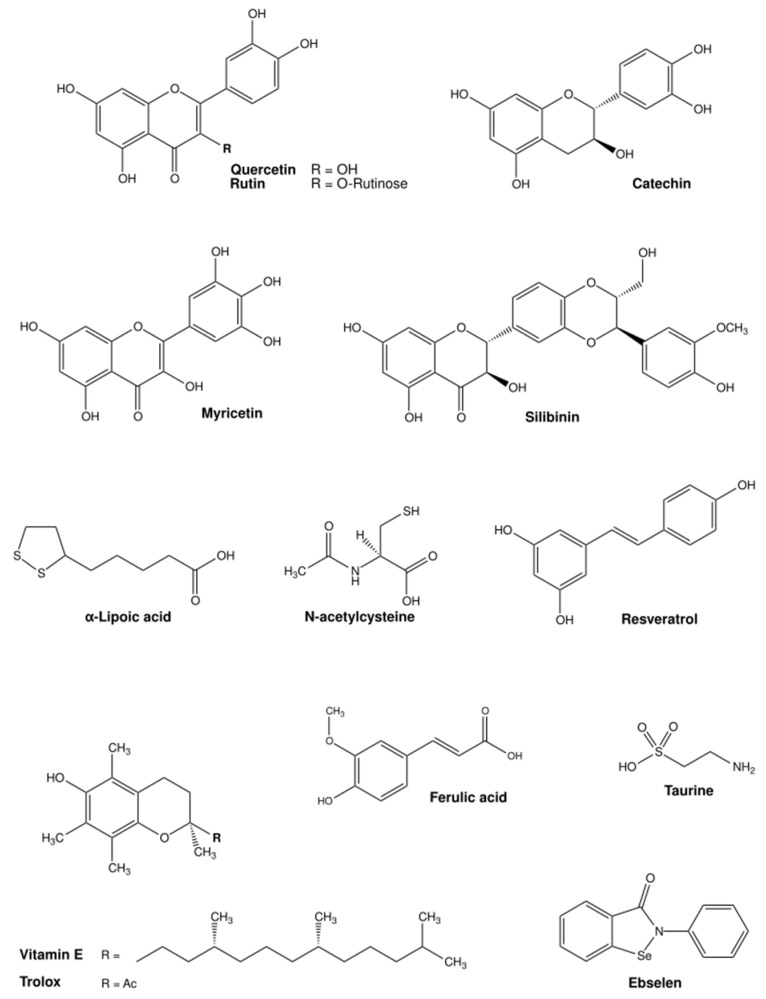
Chemical structures of commonly used antioxidants from (pre-)clinical trials for the treatment of metal toxicity. Structures were illustrated with ChemDraw Professional, Revity Signals Software. Sources: quercetin [184], rutin [185], catechin [186], myricetin [187], silibinin [188], α-lipoic acid [189], N-acetylcysteine [190], resveratrol [191], ferulic acid [192], taurine [193], vitamin E [194], Trolox [195], and ebselen [196].

**Table 1 ijms-25-07880-t001:** Antioxidant treatments in pre-clinical and clinical models to ameliorate iron, copper, and manganese metal toxicity.

Iron		Copper		Manganese	
Compound	Effects	Compound	Effects	Compound	Effects
Ebselen (in vitro)	↓ Fe-uptake ↓ ROS	Resveratrol, Ferulic acid, Phloretin Tetrahydro curcuminoids (in vitro)	↑ Cell viability and proliferation ↓ ROS	Silymarin (in vitro, co-treat)	↓ H_2_O_2_ and ROS levels ↑ MnSOD, SOD, GPx, CAT activity
Quercetin Catechin Rutin (in vitro)	↓ Hemolyzed GSH ↓ MDA levels, CAT activity	Pitaya-extract (in vivo)	↓ AChE activity ↓ Lipid peroxidation ↓ MDA, CAT levels	Açaí extract (anthocyanins) (in vitro)	↑ GSH/GSSG ratio ↑ Glutamate uptake ↓ Lipid peroxidation ↓ Nrf2 activation
α-Lipoic acid + vitamin D (in vitro)	↓ Fe-content ↓ ROS ↓ p53, APP and SOD content	Pitaya-extract (in vivo, co-treatment)	↓ GST, CAT activity (brain) ↑ SOD activity (brain) ↑ GST activity (gut) ↓ CAT activity (gut) ↑ AChE and δ-aminolevulinate dehydratase activity (brain) ↓ cortisol levels	N-acetylcysteine/GSH (in vitro, pre-treatment)	↑ Cell viability ↓ DNA damage
Myricetin (in vitro, in vivo)	↓ Fe-uptake ↓ TfR1 ↓ Lipid peroxidation ↓ DNA oxidation products	Vitamin E (in vivo, co-treatment)	↑ GSH, GPx levels ↓ H_2_O_2_, MDA levels	Quercetin (in vivo)	↓ H_2_O_2_ levels ↓ Lipid peroxidation ↑ SOD, CAT activity ↓ AChE activity ↑ Locomotor impairment
Taurine (in vivo)	↓ Lipid peroxidation ↓ Loss of GSH levels	Silymarin (in vivo, encapsulated in a liposomal formulation)	↑ Spatial memory ↓ Liver damage (AST, ALT, Total-Bilirubin)	Quercetin (in vitro, pre-treatment)	↑ Cell viability ↓ ROS and MDA levels ↑ SOD, CAT, GSH activity ↓ Loss of MMP ↓ TNF-α, IL-1β, IL-6 protein content ↓ NF-kB, iNOS mRNA levels ↑ HO-1, Nrf2 mRNA levels ↓ Bax, Cyt c, caspase 3, PARP-1 levels
Quercetin (in vivo, human studies)	↑ Fe-depletion ↓ Oxidative stress ↓ C-reactive protein levels ↓ Ferritin content ↓ Transferrin saturation ↑ Transferrin levels	α-Lipoic acid (in vivo)	↓ Liver damage ↑ GPx, GR, SOD activity ↓ Lipid peroxidation	Lycopene (in vivo, pre-treatment)	↓ ALT, AST, AChE, glucose levels (serum) ↓ AChE levels (brain)
Silymarin (human studies)	↓ Fe-content ↓ Ferritin content ↓ Fe-binding capacity ↓ Hepcidin and transferrin levels	N-acetylcysteine (in vivo)	↓ Cu/Zn ratio ↓ Liver and kidney damage	Trolox (in vivo, co-treatment)	↓ Caspase activity ↓ Oxidative stress ↑ Motor coordination deficits
				Trolox (in vitro/in vivo, pre-treatment)	↓ Oxidative stress ↓ ATP depletion
↑ increase, ↓ decrease					

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
