# Peer review of "Metabolic Derangement of Essential Transition Metals and Potential Antioxidant Therapies"

_ijms, 2024, doi:10.3390/ijms25147880_

Round 1
Reviewer 1 Report
Comments and Suggestions for Authors
The manuscript "Metabolic derangement of essential transition metals and potential antioxidant therapies" is an interesting review of the physiological roles of transition metal cations.
The manuscript summarizes the role of the ions Zn2+, Cu2+/1+, Fe+3/+2, and Mn+7/+4/+2.
Despite the popular language used mainly to refer to the iron, zinc, manganese, and copper supplements, the biochemistry-physiological education and science must refer to the respective ions (cations) of metals once the metallic state is absent in vivo.
Thus, the manuscript needs extensive revision to refer to the ions (Zn2+, Cu2+/1+, Fe+3/+2, and Mn+7/+4/+2) instead of referring to Fe, Zn, Cu, and Zn throughout the text.
The other point is the lack of figures for the topics. Can the authors try to illustrate the text better? Figure 1 needs to be edited to show the charge of the cations and also to improve the resolution.
Author Response
Comment 1: The manuscript "Metabolic derangement of essential transition metals and potential antioxidant therapies" is an interesting review of the physiological roles of transition metal cations. The manuscript summarizes the role of the ions Zn2+, Cu2+/1+, Fe+3/+2, and Mn+7/+4/+2. Despite the popular language used mainly to refer to the iron, zinc, manganese, and copper supplements, the biochemistry-physiological education and science must refer to the respective ions (cations) of metals once the metallic state is absent in vivo. Thus, the manuscript needs extensive revision to refer to the ions (Zn2+, Cu2+/1+, Fe+3/+2, and Mn+7/+4/+2) instead of referring to Fe, Zn, Cu, and Zn throughout the text.
Reply 1: We thank the comment of the reviewer. To address it, the authors carefully revised the manuscript to include the oxidation state of the metals in the text, and also in the figures.
Comment 2: The other point is the lack of figures for the topics. Can the authors try to illustrate the text better? Figure 1 needs to be edited to show the charge of the cations and also to improve the resolution.
Reply 2: We thank the comment and the suggestion of the reviewer. The manuscript was extensively revised to refer to the ions with their respective oxidative state in the biological system. Furthermore, the same revision was applied to Figure 1 (renamed to Figure 5 in the revised manuscript). The text was further ilustrated with Figures 1,2,3,4 and 6.
Reviewer 2 Report
Comments and Suggestions for Authors Dear authors,After reviewing the following manuscript entitled "Metabolic derangement of essential transition metals and potential antioxidant therapies” (ijms - 3042147), I sent the following comments and observations that the authors should attend to before its publication in this journal.
Since it is a review, I suggest making a prisma flow for documentation as well as presenting the work methodology in terms of documentation.
The authors must mention which data collection method he used. It should be mentioned from the databases used and what is the situation of the articles that have data of interest to the authors under study.
Figure 1 is not clear in appearance.
It is a very old bibliography. I propose to remove references older than 10 years.
Author Response
Comment 1: Since it is a review, I suggest making a prisma flow for documentation as well as presenting the work methodology in terms of documentation. The authors must mention which data collection method he used. It should be mentioned from the databases used and what is the situation of the articles that have data of interest to the authors under study.
Reply 1: We thank the comment of the reviewer. However, since our manuscript is neither a systematic review nor a meta-analysis, we consider that a Prisma flow would not fit in this specific case. The focus of this review was to give an overview of the cellular metabolism of transition metals, consequences of their overload and current therapies, followed by their potential role in inducing oxidative stress upon overload and remedies thereof. To this, we included a broad range of bibliographic sources, from other reviews to pre-clinical and clinical evidence.
Comment 2: Figure 1 is not clear in appearance.
Reply 2: We thank the comment of the reviewer. Figure 1 (renamed to Figure 5 in the revised manuscript) was provided with a better resolution.
Comment 3: It is a very old bibliography. I propose to remove references older than 10 years.
Reply 3: We thank the comment of the reviewer. The manuscript was subjected to an extensive revision of the bibliography. The references older than 10 years that were kept in the revised manuscript are only the ones considered of high relevance to this publication. Furthermore, more recent references were added when appropriate, aiming to improve the quality of the manuscript.
Round 2
Reviewer 1 Report
Comments and Suggestions for Authors
All issues have been addressed and the manuscript can now be accepted for publication.